# A Three-Protein Panel to Support the Diagnosis of Sepsis in Children

**DOI:** 10.3390/jcm11061563

**Published:** 2022-03-12

**Authors:** Francisco J. Pilar-Orive, Itziar Astigarraga, Mikel Azkargorta, Felix Elortza, Susana Garcia-Obregon

**Affiliations:** 1Pediatric Critical Care Group, Biocruces Bizkaia Health Research Institute, 48903 Barakaldo, Spain; 2Pediatric Critical Care Service, Hospital Universitario Cruces, 48903 Barakaldo, Spain; 3Pediatric Oncology Group, Biocruces Bizkaia Health Research Institute, 48903 Barakaldo, Spain; itziar.astigarraga@osakidetza.eus; 4Pediatric Service, Hospital Universitario Cruces, 48903 Barakaldo, Spain; 5Pediatric Department, Universidad del País Vasco UPV/EHU, 48940 Leioa, Spain; 6Proteomics Platform, CIC bioGUNE, Basque Research and Technology Alliance (BRTA), CIBERehd, ProteoRed-ISCIII, Bizkaia Science and Technology Park, 48160 Derio, Spain; mazkargorta@cicbiogune.es (M.A.); felortza@cicbiogune.es (F.E.); 7Physiology Department, Universidad del País Vasco UPV/EHU, 48940 Leioa, Spain

**Keywords:** mass spectrometry analysis, proteome, sepsis, septic shock, children, biomarkers

## Abstract

Sepsis is a syndrome without a standard validated diagnostic test. Early recognition is crucial. Serum proteome analysis in children with sepsis may identify new biomarkers. This study aimed to find suitable blood biomarkers for an early diagnosis of sepsis. An analytical observational case-control study was carried out in a single center. Children admitted to a Pediatric Intensive Care Unit with clinical diagnosed sepsis were eligible for study. A proteomic analysis conducted by mass spectrometry was performed. Forty patients with sepsis and 24 healthy donors were recruited. Proteomics results revealed 44 proteins differentially expressed between patients and healthy controls. Six proteins were selected to be validated: lactoferrin, serum amyloid-A1 (SAA-1), complement factor B, leucine-rich alpha-2 glycoprotein (LRG1), soluble interleukin-2 alpha chain receptor (sCD25) and soluble haptoglobin–hemoglobin receptor. Our results showed that sCD25, SAA-1, and LRG1 had high levels of specificity and sensitivity, as well as an excellent area under the ROC curve (>0.9). Our study provides a serum proteomic analysis that identifies new diagnostic biomarkers in sepsis. SAA-1, sCD25 and LRG1 were able to separate septic from healthy donor, so they could be used together with other clinical and analytical features to improve sepsis diagnosis in children.

## 1. Introduction

Sepsis is defined as a life-threatening organ dysfunction caused by a dysregulated host response to infection and “septic shock” defined as a subset of sepsis with circulatory and cellular/metabolic dysfunction associated with a higher risk of mortality [1]. Although there have been attempts to apply the definition of Sepsis-3 to children with promising results [2], most physicians still use the definition of the Consensus Conference on Pediatric Sepsis published in 2005 [3,4].

Sepsis in children is one of the main causes of morbidity, mortality and consumption of health resources worldwide. The estimated number of cases is 48 sepsis and 22 severe sepsis cases in children per 100,000 person-years. The population-level estimate for neonatal sepsis is 2202 per 100,000 live births. Extrapolating these figures on a global scale, we estimate an incidence of 3.0 million cases of sepsis in neonates and 1.2 million cases in children [5]. Mortality rate ranges from 1% to 20%, depending on the severity of the disease, comorbidities and geographical location [6,7]. Most children die because of refractory shock and multiorgan dysfunction syndrome. Most deaths occur within the first 48–72 h of onset [8]. A crucial aspect to improve the outcomes in sepsis is the early recognition based on clinical conditions (Pediatric Early Warning score (PEWS), Sequential Organ Failure Assessment (SOFA score), quick SOFA) and/or biomarkers, allowing prompt treatment to be started. Sepsis is diagnosed by clinical and microbiological test, but unfortunately, gold standard diagnostic tests are not currently available and microbiological results could take some days. Several biomarkers may predict serious bacterial infections. Amongst them, procalcitonin (PCT), and C-reactive protein (CRP) are the most used, but unfortunately, they have not been shown to have good specificity and sensitivity and none of them has achieved the goals of a good biomarker for sepsis [9].

Analysis of specific protein patterns in biofluids is relevant for both the understanding of pathogenesis and the definition of disease markers. Among biological fluids, serum is likely to be the best candidate [10]. Analysis of the serum proteome from children with sepsis may identify new biomarkers related to disease development and progression [11,12]. The objective of this study is the identification of new serum biomarkers for the diagnosis of sepsis in children that will allow recognition for the early identification and treatment of the disease.

## 2. Materials and Methods

### 2.1. Study Population: Inclusion and Exclusion Criteria

An analytical observational case-control study was carried out at Hospital Universitario Cruces (Bizkaia, Spain) from March 2013 to October 2016.

Pediatric patients (age from 1 month to 16 years) admitted to PICU with a diagnosis of severe sepsis/septic shock according to the Consensus Conference on Pediatric Sepsis 2005 [3,4] were enrolled in this study. Patients who had recently received blood products or those with immunosuppressed conditions or with other immunological/inflammatory diseases were excluded.

Controls were healthy children who required blood tests as part of a preoperative study for minor surgery (groin hernia, etc.) or those without proven disease (constitutional growth delay, etc.). Patients were not considered for controls if they had a fever or an acute inflammatory or inflammation disease.

### 2.2. Data and Sample Collection

Demographics and clinical data (age, sex, weight, place of admission, personal history, organ dysfunction, etc.); vital signs and laboratory tests, (CRP, PCT and lactate); were collected. Microbiological data, general management, outcomes, severity scores (PRISM III), length of stay in PICU and hospital, were also recorded. The study was performed according to Spanish Law (Biomedical Research and Protection of Personal Data) and Declaration of Helsinki.

Blood collection in septic patients was taken in the PICU, through a peripheral or central venous line, at the time of admission. For healthy donors, the samples were collected in the blood draw area at the hospital through the same venous access used for blood tests requested for preoperative study. Samples were allowed to clot at room temperature and were centrifuged at 2000 rpm for 20 min. Separated serum was stored at −80 ºC in the Basque Biobank for Research-OEHUN until analysis. Extraction and sample process were made based on a standardized protocol.

### 2.3. Proteomic Study

For the proteomic analysis, 15 serum samples were selected to work with in this study: ten from septic patients (five Gram + and five Gram-) and five from healthy donors. The analysis of serum proteome seems to be challenging due to the difficulty in finding low-abundance target proteins because of the presence of high serum proteins (albumin and immunoglobulin, among others) [11]. Thus, a kit was used to deplete those abundant proteins and enhance the detection of lower abundant proteins. Thermo Scientific Pierce Top 12 Abundant Protein Depletion Spin Columns (Thermo Scientific, Walthman, MA, USA) were used for single-step removal of twelve high-abundance proteins from 10 μL of serum samples. Samples were digested following the FASP protocol described by Wisniewski et al. [13]. Nano scale liquid chromatography coupled on-line to tandem mass spectrometry was performed for protein identification and relative quantification (nLC MS/MS). LC was performed using a NanoAcquity nano-HPLC (Waters), and LTQ Orbitrap XL (Thermo Scientific) mass spectrometer was used for the analysis. Database searching was performed using MASCOT 2.2.07 (Matrixscience, London, UK) against a UNI-PROT–Swissprot database filled only with entries corresponding to Homo sapiens (without isoforms). Progenesis LC-MS (version 2.0.5556.29015, Nonlinear Dynamics) was used for the label-free differential protein expression analysis. The significance of expression changes was tested at protein level, and proteins identified with at least two peptides and an ANOVA *p*-value ≤ 0.05 and a ratio > 2 in either direction were selected for further analyses. All samples were analyzed for each marker as a single batch.

### 2.4. Selection of Proteins and Their Validation by ELISA

All the above mentioned proteins were submitted to STRING program (STRING: functional protein association networks. https://string-db.org/ (accessed on 31 December 2021)), to identify the biological processes in which these proteins were involved and to select those related to the innate and adaptive immune system and the inflammatory response (immune and inflammatory response, activation of coagulation and complement pathway). After this, a bibliographic review was conducted in Pubmed database for selecting proteins for validation. Finally, the selected proteins were validated by sandwich Enzyme-Linked Immunosorbent Assay (ELISA) in the cohort of patients and controls. In all cases, ELISA assays were performed using commercial kits and following the manufacturer’s procedures (R&D Systems, Minneapolis, MN, USA for sCD25 and sCD163 and Cloud-Clone Corp. CCC, USA for LRG1, LTF, CFAB and SAA-1).

### 2.5. Statistical Analysis

Data were reported using mean (standard deviation) for the symmetrically distributed variables and median (first quartile–third quartile) for variables that showed an asymmetrical distribution. The Kolmogorov–Smirnov test was used to evaluate if the variables followed a normal distribution. To evaluate the differences between variables we used non-parametric tests (Mann–Whitney U) if the variables did not follow a normal distribution and Student’s *t*-test for those with a normal distribution. A *p* value < 0.05 was considered statistically significant. Bivariate correlations of the proteins with biomarkers used in clinical practice were performed using the Pearson correlation coefficient.

The receiver operating characteristic (ROC) curves and areas under the curve (AUC) determined the variable with the greatest predictive value, with 95% confidence intervals [95% CI] compared to determine significance. ROC and AUC were calculated for each statistically significant protein. Youden’s criterion was used to establish optimal cut-off values.

Statistical analysis including demographic, clinical and laboratory variables was per-formed using SPSS 23 (SPSS Inc., Chicago, IL, USA) for Windows.

## 3. Results

During the period of study 64 samples were collected: 40 septic patients and 24 healthy controls. The control healthy group included 12 males (50%) and 12 females (50%), with a median age of 8.2 years (P25-P75: 3.71, 12.44), range 1.27–15.68 years.

The demographic of the patients with sepsis are shown in Table 1. Seventy per cent of sepsis cases were admitted from the Emergency Department. The distribution of subjects with sepsis and septic shock is also provided in Table 1. The prevalence of sepsis in the Pediatric Intensive Care Unit (PICU) population was 3%, with septic shock accounting for 82.5% of the patients with sepsis. According to other studies on sepsis, 21 of the patients with sepsis (52%) had positive blood cultures [14]. The most frequent source of infection was endovascular (57%), followed by pneumonia. Meningococcus (57%), Streptococcus pyogenes (19%) and Streptococcus pneumoniae (9.5%) were the most common microorganisms involved. Laboratory analysis for C-reactive protein (CRP), Procalcitonin (PCT) and lactate were carried out as part of the routine studies performed in patients with suspected severe infection. CRP was negative in seven patients (17.5%), three of them with a confirmed bacterial infection; PCT was negative in four patients (10%), none of them had a confirmed bacterial infection. Twenty-three patients reported high lactate levels on admission (57.5%), compared with 17 who had normal values. The mortality rate was 2.5%.

### 3.1. Proteomic Results

Label-free relative quantification proteomic analysis on 15 serum samples, resulted in 232 differentially expressed proteins. After applying the selection criteria described in methods, only 44 proteins were significantly deregulated between healthy donors and patients (Appendix A). Additionally, we found 58 and 59 proteins significantly deregulated between Gram+ and healthy donors and between Gram− and healthy donors, respectively. There were 36 common proteins among three groups (Figure 1).

### 3.2. Selection of Proteins and Their Validation by ELISA

Among these 36 proteins, 24 were directly involved in processes related to the host’s immune response according to STRING program. Those proteins previously studied which had not proved to be significantly sensitive and specific were removed from the study.

After performing a bibliographic search, we selected the following proteins to be validated: Lactotransferrin (LTF), Complement Factor B protein (CFB), Serum Amyloid A-1 (SAA-1), Leucine-rich alpha-2-glycoprotein (LRG1) (see Figure 2). Two more proteins were added to our study, the soluble interleukin-2 receptor alpha chain (sCD25) and the soluble haptoglobin-hemoglobin receptor (sCD163). These two last proteins are used as biomarkers in the diagnosis and monitoring of hemophagocytic lymphohistiocytosis (HLH) activity [15], as an example of inflammatory disease with immune dysregulation and exaggerated inflammatory response [16], so they could be interesting to be tested in sepsis.

The validation results of the six previous mentioned significantly deregulated protein performed by ELISA are shown in Table 2. SAA-1 (*p* < 0.001), sCD25 (*p* < 0.001), LRG1 (*p* < 0.001) and LTF (*p* < 0.001) serum concentration proteins showed statistically significant differences between septic patients and healthy controls. ROC curves for each protein were determined to estimate sensitivity and specificity as a biomarker in the diagnostic of septic patients (severe sepsis and septic shock) and data could be seen in Table 2.

The cut-off values of each protein obtained using the Youden index were as follows: SAA-1: 35,891 ng/mL, sCD25: 3209 pg/mL, LRG1: 56,951.5 ng/mL, LTF: 16,349 ng/mL, sCD163: 1123 pg/mL and CFAB: 296,919 ng/mL.

Next, we performed bivariate correlations among the different biomarkers: Lactate, PCT, CRP, sCD25, CD163, SAA-1, CFB, LTF and LRG1. A positive correlation was observed between SAA-1 and LTF protein (r = 0.52) and between SAA-1 and CRP (r = 0.412). There was also a negative correlation between PCT and CFB (r = 0.488).

Among these six proteins studied as potential biomarkers, three of them, SAA-1 (AUROC 0.978; 95% CI: 0.946–1), sCD25 (0.97; 95% CI: 0.92–1) and LRG1 (0.93; 95% CI: 0.86–1), showed an area under the ROC curve greater than 0.9. In Figure 3, the area under the receiver operating characteristic (AUROC) and box plot for these proteins are shown.

All septic patients were positive for at least one of these three proteins (SAA-1, sCD25 and LRG1) and all healthy donors were negative, which allowed us to split up all the patients and control by analysis these three proteins, remarking the value of proteins in the diagnosis of sepsis.

## 4. Discussion

In our study, based on the analysis of the serum proteome of septic patients, we found a three-protein signature that might facilitate the diagnosis of sepsis and allow an earlier start of treatment. To our best knowledge, several sepsis related clinical proteomic studies have been carried out by mass spectrometry showing a very broad range of deregulated proteins [17] but they failed in identifying proteins easy to perform in clinics.

Pierrakos et al., in a recent review on biomarkers in sepsis, identified 258 possible candidates. The authors illustrated that, although new biomarkers have been proposed, little progress has been made in identifying those with clinical relevance [18].

To date, none of the identified biomarkers has been sufficiently specific and sensitive to be considered as a “gold standard”. Research has been focused on individual biomarkers such as interleukin 6, the surface receptor expressed on myeloid cells (sTREM1), proadrenomedulin (MR-proADM), lactate, CRP and PCT, among others. In sepsis the most representative are PCT, CRP and lactate. A recent meta-analysis of 17 studies, included 1408 patients (1086 neonates and 322 children) with early and late onset sepsis, identified a sensitivity of 0.85 and specificity of 0.54 at the PCT cut-off of 2.0–2.5 ng/mL [19]. Diaz et al. found that a CRP cutoff greater than 3 mg/dL did not identify more than 80% of infants with invasive bacterial infection [20]. Furthermore, CRP, PCT and lactate biomarkers have shown important deficiencies that make them unreliable for this purpose. As we have shown in our study, the failure rate of these biomarkers is relatively high, since they did not allow us to split up patients and healthy donors.

However, in our analysis, we identified a number of potential proteins for differentiation between septic patients and healthy controls. Among the four proteins that selected to validation by ELISA, two of them (SAA-1 and LRG1) showed high values of sensitivity and specificity. These proteins have been described separately in sepsis by some authors.

Arnon et al., in their study about sepsis in infants, compared CRP with SAA-1. They found that SAA-1 levels increased earlier and higher, returning to normal values more quickly than CRP. SAA-1 was more accurate in predicting early-onset sepsis than CRP (sensitivity (96 vs. 30%), specificity (95 vs. 98%) [21]. Our results are in concordance with this study.

Hashida et al., performed a proteome analysis of hemofilter adsorbates from patients with sepsis and identified 197 proteins. Three proteins, including carbonic anhydrase 1 (CA1), LRG1 and cystatin-C (CysC) were present in all samples from septic patients, a fact that had not been previously reported in septic patients [22]. Validation analysis of patients’ serum revealed that patients with sepsis had increased serum levels of CA1 and LRG1 compared to patients without sepsis (*p* < 0.05). There was a non-significant increasing trend in serum CysC levels. Subsequent studies have shown that LRG1 expression increased in hepatocytes in response to mediators of the acute-phase response, and serum LRG1 levels were increased in patients with bacterial infections [23]. LRG1 is also present in the peroxidase-negative granules of human neutrophils. Neutrophils are traditionally considered as the first responders of the innate immune system, owing to their intrinsic capacity to eliminate pathogenic organisms [24].

The third biomarker validate in our analysis, sCD25 or the soluble IL2 receptor alpha chain, showed a good sensitivity and specificity. sCD25 has been found to be higher in patients with sepsis than in patients with non-infective SIRS. sCD25 performed well as biomarkers of sepsis irrespective of the severity of illness. The area under the receiver operating curve (ROC) was 0.902 (95% CI: 0.854–0.949). Therefore, this biomarker has better performance than PCT to identify sepsis at ICU admission, as has been previously published [25].

The soluble interleukin-2 receptor alpha chain (sCD25) and the soluble haptoglobin-hemoglobin receptor complex (sCD163) have been used as biomarkers in the diagnosis and surveillance of hemophagocytic lymphohistiocytosis (HLH) activity. Several studies suggest that both sCD163 and sCD25 might be promising markers of sepsis [25,26]. In a previous proteomics study conducted by our group in adult sepsis patients, we proposed a set of 10 proteins (AT-III, CLUS, SAA-1, HO-1, IL-6, IL-18, sCD25, sCD163, sICAM-1 and sFas) that can be considered as a complementary tool for the clinicians in the diagnosis of sepsis [27]. Two of them, SAA-1 and sCD25, have also been found to be excellent markers of infection in children in the present study.

Serum levels of sCD25 replicate the level of CD25 expression on activated T cells, which has been suggested as a marker of activation-induced regulatory T cell response [28]. Whereas levels of acute phase reactants, such as PCT and CRP, reflect the magnitude of the inflammatory response, expression of CD25 may reproduce the development of a compensatory anti-inflammatory state and thus provide additional information about an individual’s response to sepsis at a point in time.

Our findings are in line with Saito et al. who found higher levels of sCD25 in 20 septic patients compared to 16 patients with non-infective SIRS [28]. More recently, Lvovschi et al. performed multiplex cytokine analysis on 126 patients presenting to the emergency department with non-infective SIRS or sepsis [29]. Although they were unable to demonstrate profiles characteristic of sepsis, sCD25 was the only marker independently associated with severe sepsis in multivariate analysis.

Due to the limitations of the biomarkers when analyzed individually, it has been proposed to make combinations of them (biomarker panel) to cover the different aspects of the host’s response. The combination of several biomarkers has the theoretical advantage of improving diagnostic accuracy and clinical utility.

It is not clear from the existing literature whether the biomarkers included in such panels should be selected based on pathophysiological or other criteria. The combination of a biomarker panel with clinical information may be particularly helpful in the diagnosis of sepsis [30,31]. The Pediatric Sepsis Biomarker Risk Model (PERSEVERE) study developed by Wong et al. [32] uses a combination of five different serum biomarkers to provide a risk stratification mechanism with respect to mortality and disease severity. We believe that the combination of non-related proteins involved in different pathways can improve diagnostic accuracy. 

The selected protins in our panel of biomarkers are quite representative of the different biological processes of the response to infections. They include acute phase proteins (SAA-1), immunosuppressive phase proteins (sCD25) and cell activation proteins (LRG1). By including proteins from different phases of the response to infection, their diagnostic value might be increased.

Despite the fact that our study has several limitations such as small population, low availability of the ELISA test for clinics and not validation of all the identified candidate biomarkers for sepsis by proteomics, we successfully validated from a research point of view, a three promising protein panel (SAA-1, LRG1 and sCD25) which could be performed at clinics and may help in the early state of diagnosis of sepsis in children.

## 5. Conclusions

In our label-free based relative quantification serum proteomic analysis, we detected 232 differentially regulated proteins between septic patients and healthy donors. After a rather conservative selection, 44 were considered for further analysis. Among these, serum amyloid A-1 and leucin-2-rich alpha-2 glycoprotein, together with the soluble interleukin-2 alpha chain receptor, were selected as candidates for a sepsis biomarker panel since all septic patients were positive for at least one of these three proteins and all controls were negative.

From a practical point of view, validation of these results in a large cohort of patients is needed but the introduction of a protein panel composed of SAA-1, LRG 1 and sCD25 to be used in Emergency Room protocols might show their utility and effectiveness in clinics for early recognition and treatment of sepsis in children.

## Figures and Tables

**Figure 1 jcm-11-01563-f001:**
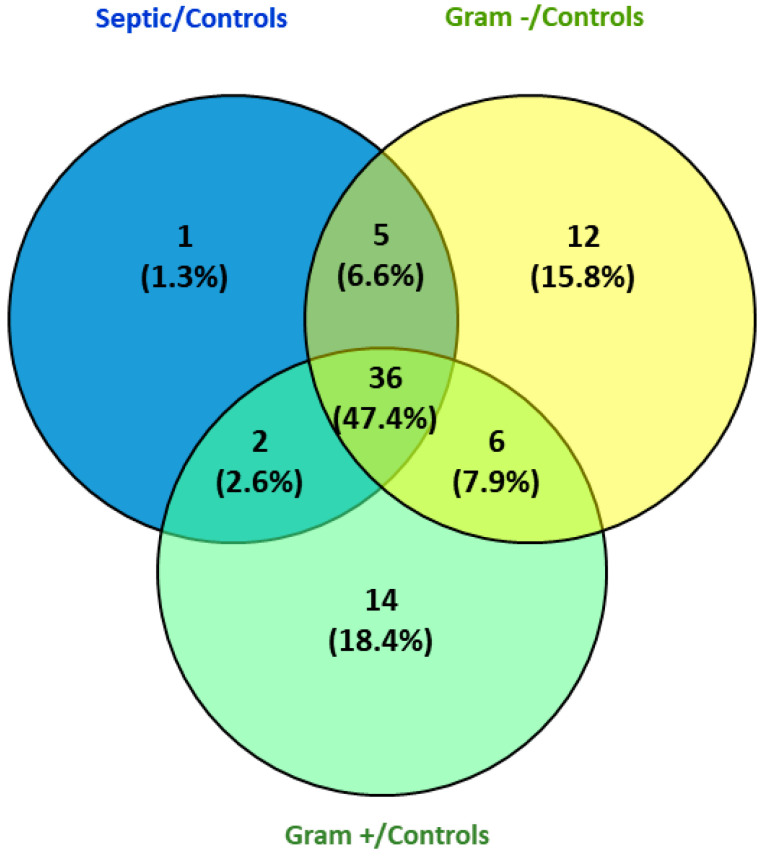
Significantly deregulated proteins among different groups.

**Figure 2 jcm-11-01563-f002:**
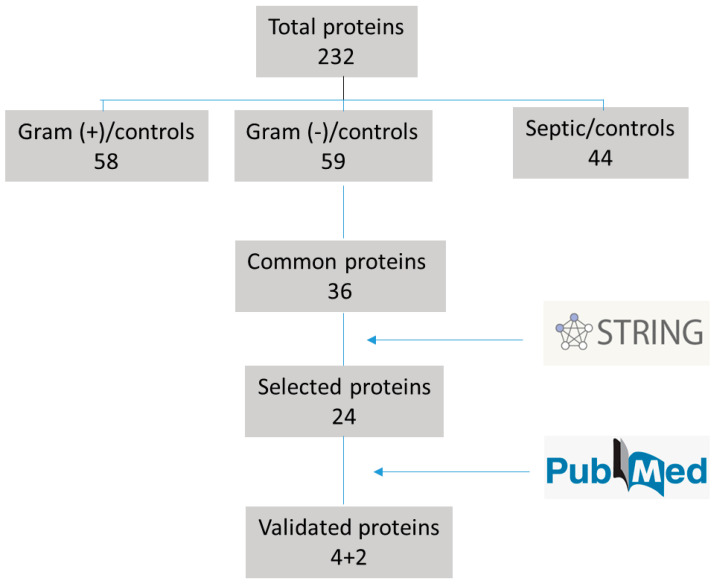
Flow chart performed for deciding validation of proteins.

**Figure 3 jcm-11-01563-f003:**
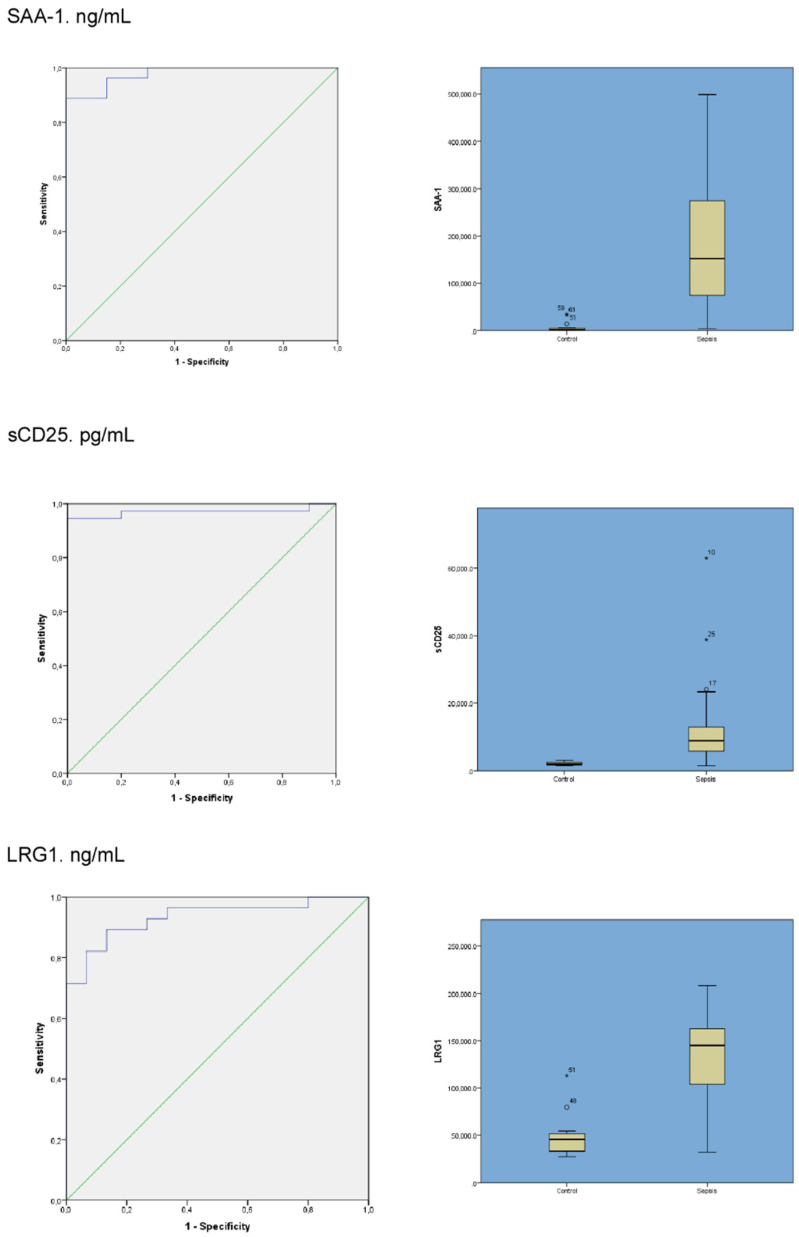
Area under the receiver operating characteristic (AUROC) and box plot for SAA-1, sCD25 and LRG1. Among these proteins studied as potential biomarkers, SAA-1 (AUROC 0.978; 95% CI: 0.946–1), sCD25 (0.97; 95% CI: 0.92–1) and LRG1 (0.93; 95% CI: 0.86–1), showed an area under the ROC curve greater than 0.9.

**Table 1 jcm-11-01563-t001:** Clinical features of patients diagnosed with severe sepsis and septic shock.

	Patients (*n*= 40)	Severe Sepsis (*n* = 7)	Septic Shock (*n* = 33)
**Gender (% female)**	18/22 (55%)	5/2 (71.4%)	17/16 (51.5%)
**Age (in years) (median P_25_–P_75_)**	3.83 (1.6–8.3)	1.04 (0.38–8.7)	4.02 (2.28–8.5)
**Weight (kg) (median P_25_–P_75_)**	16 (11–29)	8.7 (7–35)	17 (12–29)
**PRISM (median P_25_–P_75_)**	7 (4–11)	4 (2.7–4.5)	8 (4.5–13.5)
**Comorbidities:**			
Prematurity	3 (7.5%)		1 (3%)
Heart Disease	3 (7.5%)	2 (28.6%)	3 (9%)
Chronic respiratory failure	4 (10%)		3 (9%)
Neurological disease	4 (10%)	1 (14.3%)	4 (12%)
**Origin**			
Emergency	28 (70%)		24 (72.7%)
Ward	2 (5%)	4 (57.1%)	2 (6.1%)
Other hospital	9 (22.5%)		6 (18.2%)
Theatre room	1 (2.5%)	3 (42.9%)	1 (3%)
**Admission**			
Medical	38 (95%)	7 (100%)	31(94%)
Surgical	2 (2.5%)		2 (6%)
**Organ dysfunction**			
Cardiovascular	33 (82.5%)		33 (100%)
Respiratory	12 (30%)		10 (30%)
Renal	8 (20%)	2 (28.6%)	8 (24%)
Coagulopathy	7 (17.5%)		7 (21%)
Neurological	5 (12.5%)		5 (15%)
Liver	3 (7.5%)		3 (9%)
Thrombocytopenia	2 (5%)		2 (6%)
**Infection**			
Microbiologically proven	21 (52%)	4 (57.1%)	17 (51.5%)
**Site of infection**			
Endovascular	21 (52.5%)	2 (28.6%)	19 (57.6%)
Pneumonia	8 (20%)	5 (71.4%)	3 (9%)
Intra-abdominal	3 (7.5%)		3 (9%)
Others	8 (20%)		8 (18.3%)
**Treatment at PICU**			
Vasoactive *n* (%)	33 (82.5%)		33 (100%)
**NE *n*, dose µg/k/min**	16 (0.3; 0.05–1)		16 (0.3; 0.05–1)
**EPI *n*, dose µg/k/min**	14 (0.3; 0.1–1)		14 (0.3; 0.1–1)
**DOP *n*, dose µg/k/min**	26 (10; 5–20)		26 (10; 5–20)
**MV *n* (%)**	13 (32.5%)	1 (14.3%)	12 (36.4%)
**NIV *n* (%)**	1 (2.5%)	1 (14.3%)	
**AKI *n* (%)**	7 (17.5%)		7 (21%)
Hydrocortisone *n* (%)	11 (27.5%)	1 (14.3%)	10 (30.3%)
Insulin *n* (%)	1 (2.5%)		1 (3%)
**Length of support (d)**			
Vasoactives (median P_25_–P_75_)	2 (1–3)		2 (1–3)
MV (median P_25_–P_75_)	3 (2–5)	12	3 (2–5)
**Length of stay**			
PICU	3 (2–6)	2 (1–5)	3 (2–6)
Hospital	7 (6–9.5)	7 (6–14)	7 (6–9)
**Outcome**			
Survivors	39	7	32
Non-survivors	1		1

PRISM: Pediatric Risk of Mortality; NE: Norepinephrine, EPI: Epinephrine, DOP: Dopamine; MV: Mechanical ventilation; NIV: Noninvasive ventilation; AKI: Acute kidney injury. *n*: number.

**Table 2 jcm-11-01563-t002:** ELISA-validated serum protein concentrations, sensitivity and specificity, and area under the curve (AUROC) for each protein.

Protein (Units)	Sepsis (Mean/SD)	Control (Mean/SD)	*p*-Value	Sensitivity	Specificity	Area Under the Curve (AUROC) (95% CI) (DeLong)
SAA-1 ng/mL	188,472.7 ± 148,147.6	5965.7 ± 10,308.8	<0.001	0.889	1	0.978 (0.946–1)
sCD25 pg/mL	11,886.1 ± 11,334.7	2119.1 ± 500.5	<0.001	0.946	1	0.97 (0.92–1)
LRG1 ng/mL	131,052.8 ± 46,256.1	48,119.2 ± 22,320.2	<0.001	0.867	0.89	0.933 (0.86–1)
LTF ng/mL	51,940.2 ± 71,933.6	9640.8 ± 9840.9	<0.001	0.724	0.9	0.83 (0.71–0.94)
sCD163 pg/mL	1291.4 ± 6444	915.3 ± 377	0.079	0.595	0.8	0.68 (0.51–0.85)
CFAB ng/mL	188,472.7 ± 148,147.6	316,717.6 ± 431,551.4	0.078	0.621	0.73	0.65 (0.49–0.8)

SD: standard deviation; LTF: Lactotransferrin, CFAB: Complement Factor B protein, SAA-1: Serum Amyloid A-1, LRG1: Leucine-rich alpha-2-glycoprotein; sCD25: Soluble interleukin-2 receptor alpha chain; sCD163: Soluble haptoglobin-hemoglobin receptor; CI: Confidence interval.

## Data Availability

The datasets used and/or analyzed during the current study are available from the corresponding author on reasonable request.

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
