# Peer review of "A Three-Protein Panel to Support the Diagnosis of Sepsis in Children"

_jcm, 2022, doi:10.3390/jcm11061563_

Round 1

Reviewer 1 Report

This is a very thorough attempt to identify relevant biomarkers for earlier detection of sepsis. The methodology and interpretation are clearly laid out, in particular the method of interrogating the serum for relevant biomarkers. The correlation of laboratory results with pathophysiological mechanisms adds to the relevance.

Figure 4 is not necessary - I don't feel it adds anything to the text which clearly describes the distinguishing ability of the three markers.

In the discussion, you comment on the uncertainty of measuring biomarkers on clinicopathological grounds and suggest in the conclusion that this panel should be available in the ER for point-of-care assessment. If this is the assertion, which seems valid, I would suggest that you add some content to the discussion to strengthen that conclusion. I think it is also important to add some discussion of the general availability of these tests, which could be a significant barrier to their clinical utility. If they are to be used as a front-line evaluation, it may be important to advocate for increased access.

Reviewer 2 Report

Very intriguing paper, though there are some potential areas for improvement. 

First, the methods and results sections are rather confusing, and there is a lot of redundancy (a lot of description of methods restated in the results section).  I feel it would be clearer if there was a clear description of Methods and Results in two parts: (1) the derivation process & results (the 15 patients), and (2) the validation process & results.

Other comments: when describing ROC, could you make it clear that the ROC is for a given biomarker and its correlation to sepsis.  Attached question: the ROC correlation is for sepsis AND septic shock?  Or just septic shock?

Also, I understand the derivation process for 4 of the 6 biomarkers used, but then sCD25 and sCD163 are introduced independently outside of the derivation process.  Wouldn't it be better to subject a panel of sCD markers to the same validation process?

And finally, I do understand the clarity of comparing these markers in septic vs. healthy patients, but the real "holy grail" of sepsis biomarkers would be to use them to distinguish between sepsis and other acute inflammatory conditions and non-septic infections.  Do you anticipate undertaking these types of analyses to determine whether your markers are truly specific for sepsis or just markers of acute infection or immune activation/inflammation of early phase sepsis?

Round 2

Reviewer 2 Report

An improved version, easier to follow re: methods and results.

Still, the more important (future) study would be to test these biomarkers for their ability to distinguish between septic and non-septic infected children, or sepsis and other inflammatory conditions.  This is where this tool would be far more useful!